# Twenty Years of Cerebral Ultrasound Perfusion Imaging—Is the Best yet to Come?

**DOI:** 10.3390/jcm9030816

**Published:** 2020-03-17

**Authors:** Jens Eyding, Christian Fung, Wolf-Dirk Niesen, Christos Krogias

**Affiliations:** 1Department of Neurology, Klinikum Dortmund gGmbH, Beurhausstr 40, 44137 Dortmund, Germany; 2Department of Neurology, University Hospital Knappschaftskrankenhaus, Ruhr University Bochum, 44892 Bochum, Germany; 3Department of Neurosurgery, Universityhospital, University of Freiburg, 79106 Freiburg, Germany; christian.fung@uniklinik-freiburg.de; 4Department of Neurology, Universityhospital, University of Freiburg, 79106 Freiburg, Germany; wolf-dirk.niesen@uniklinik-freiburg.de; 5Department of Neurology, St. Josef-Hospital, Ruhr University Bochum, 44791 Bochum, Germany; christos.krogias@rub.de

**Keywords:** ultrasound, acute ischemic stroke, perfusion imaging, contrast agent, intracerebral hematoma, subarachnoid hemorrhage

## Abstract

Over the past 20 years, ultrasonic cerebral perfusion imaging (UPI) has been introduced and validated applying different data acquisition and processing approaches. Clinical data were collected mainly in acute stroke patients. Some efforts were undertaken in order to compare different technical settings and validate results to gold standard perfusion imaging. This review illustrates the evolution of the method, explicating different technical aspects and milestones achieved over time. Up to date, advancements of ultrasound technology as well as data processing approaches enable semi-quantitative, gold standard proven identification of critically hypo-perfused tissue in acute stroke patients. The rapid distribution of CT perfusion over the past 10 years has limited the clinical need for UPI. However, the unexcelled advantage of mobile application raises reasonable expectations for future applications. Since the identification of intracerebral hematoma and large vessel occlusion can also be revealed by ultrasound exams, UPI is a supplementary multi-modal imaging technique with the potential of pre-hospital application. Some further applications are outlined to highlight the future potential of this underrated bedside method of microcirculatory perfusion assessment.

## 1. Introduction: Cerebral Ultrasound Perfusion Imaging (UPI), First Clinical Applications

Ultrasound imaging is a key diagnostic tool in clinical medicine. Even if an expert examiner is needed to obtain and interpret the images, it is advantageous to other diagnostic entities for various reasons, two of them being the mobile bedside character of the examination and the absence of radiation exposure. Besides gray-scale B-mode imaging for tissue characterization, vessel imaging by Doppler-based duplex-sonography is the basis in most diagnostic work-up settings. After application of specific contrast enhancing substances, improved vessel imaging and contrast-enhanced tissue imaging (CEUS) can provide sophisticated information like vascular occlusion or tissue perfusion imaging in various indications. In neurosonology, ischemic stroke and its diagnostic work up is the leading indication for ultrasound imaging questioning the vessel status of extra- and intracranial arteries. With the invention of contrast-enhanced perfusion imaging, the question of transferability to cerebral imaging quickly emerged. In 1998, the first report on the ability of tracing contrast enhancer in the cerebral microcirculation of healthy volunteers by transient harmonic imaging was published [1], followed by a case report on two acute stroke patients displaying impaired contrast increase in later infarcted areas in 1999 [2]. The technical approach was adapted from echocardiography, where size of myocardial infarct had been visualized before [3]. Various case series could reproduce the initial results by demonstrating missing signal increase in affected ischemic brain areas [4,5,6]. In the cerebral application, the temporal bone hampers ultrasound transmission resulting in relatively poor imaging quality. Therefore, different variations of harmonic imaging techniques and data acquisition and processing approaches have been introduced since to improve imaging quality [7,8,9,10]. Hereby, a novel approach displaying both hemispheres in one examination for isochronal comparison of normal and ischemic brain areas (bilateral or mirror approach) was introduced in 2003 [11] (compare “Data Acquisition and Processing” below for comparison of unilateral and bilateral approach). Using a bolus kinetic approach, time-based parameters such as TPI (time-to-peak intensity) could distinguish between areas of normal, impaired, and nullified parenchymal perfusion [12,13]. Figure 1 illustrates the conventional transversal insonation plane using the transtemporal bone window. Figure 2 illustrates an early unilateral examination of an acute stroke patient displaying missing contrast enhancement in later infarction, and Figure 3 an up-to-date bilateral examination of a normal person and an acute stroke patient displaying different areas of impaired perfusion. A systematical review of the literature on the method has recently summarized an overview until early 2017 [14].

## 2. Technical Aspects

### 2.1. Microbubbles and Harmonic Imaging

The use of ultrasound contrast enhancers (US-CE) is a prerequisite in the application of ultrasound perfusion imaging (UPI). US-CEs consist of gaseous microbubbles (diameter ranging between 1 and 10 µm), which are stabilized by various types of shells, aiming to provide high microbubble stability with improved signal-to-noise ratio and a sufficient examination time [16]. These microbubbles show strong backscattering of beamed ultrasound pulses, not only with linear scattering, but mainly with non-linear scattering, which usually is not relevantly present in most tissues. The different composition of the US-CEs that have been used so far in UPI are displayed in Table 1.

With increasing acoustic power, the microbubbles can be set into resonance vibrations, a process that results in the additional emission of harmonic frequencies—multiples of the fundamental frequency. This attribute enables various contrast harmonic imaging modes to detect the US-CE with high sensitivity and to differentiate it from the surrounding tissue. This goal is usually achieved by a band pass filter, which suppresses the fundamental frequencies.

Depending on the applied acoustic power, various interactions between the ultrasound beam and the US-CEs occur. By further increasing the ultrasound energy, the microbubbles can burst. This effect is referred to as “stimulated acoustic emission”, since bursting microbubbles emit their own ultrasound, which in turn can be used for ultrasound imaging. The mechanical index (MI), originally defined to predict the onset of cavitation in fluids, gives an on-screen indication of the likelihood of microbubble destruction during examination. MI is defined as maximum value of the peak negative pressure divided by the square root of the acoustic center frequency. The threshold between a low MI and high MI is not clearly defined in cerebral imaging; however, an MI > 1.0 is needed for the destruction of the microbubbles to compensate for the ultrasound absorption of the skull [10]. Therefore, actual acoustic intensity in brain parenchyma is far less than in other organs as expected by mere MI values because of the strong absorption of the skull. Overall, data acquisition modes can be divided in “non-destructive” and “destructive” imaging modes:

#### 2.1.1. “Non-Destructive Imaging Modes”:

Conventional Harmonic Imaging

Conventional harmonic imaging is a single pulse modality based on the described stronger non-linear oscillation of US-CEs compared to the surrounding tissue. The non-linear oscillation results in harmonic frequencies (multiples of the fundamental frequency), enabling the differentiation between the signals of tissue and microbubbles by the use of band pass filters (Figure 4).

Phase Inversion Harmonic Imaging

In phase (or pulse-) inversion harmonic imaging (PIHI), two echoes are acquired per line, resulting from a pair of mirror-inverted transmit pulses. An acoustic wave in a medium (i.e., the first transmit pulse) shows sinus-wave characteristics, so that a zone of overpressure is followed by a symmetric zone of negative pressure. In case of a linear scatterer, the summation of the two scattered and acquired echoes results in a reciprocative elimination, so that the fundamental is cancelled out. With the use of US-CEs, the non-linear oscillation changes according to the absolute pressure, so that the summation of the two echoes results in a mismatch, as the overpressure in the first echo will not be equal to the negative pressure in the second echo. This mismatch is the same for both half cycles, so that the result of the summation, in principle, is the second harmonic. Only this mismatch is visualized, so that PIHI performs the separation of the second harmonic from the fundamental [8].

Power Modulation Harmonic Imaging

Like in PIHI, power modulation harmonic imaging (PMHI) represents a further multi-pulse technique. Using multiple pulses with differences in amplitude, PMHI aims to detect the harmonic response by sending several pulses and subtracting the responses, as the linear response reduces with multi-pulsing and the harmonic response remains.

#### 2.1.2. “Destructive Imaging modes”:

Contrast Burst Imaging and Time Variance Imaging

Contrast burst imaging (CBI) and time variance imaging (TVI) are derived from Power Doppler in which pulses are broadband with high acoustic power. Power Doppler uses the Doppler shift in frequency induced by the movement of the scattering objects, displaying the amplitude of the Doppler signal, instead of displaying this frequency shift. This technique can also be combined with a harmonic bandpass filter. In this context, CBI detects the changes in the acoustic properties of microbubbles that are caused by ultrasound-induced destruction, while suppressing tissue and clutter signals by multiple echo measurements. TVI also depicts the time variant acoustic properties of microbubbles by analyzing multiple pulse echo measurements, but TVI uses a contrast-agent-specific analysis strategy to improve the suppression of noise and artifacts [9,17].

### 2.2. Data Acquisition and Processing

In order to detect the distribution of contrast enhancer in the micro vascular space, various approaches of data acquisition as well as data processing have been applied [14]. Data acquisition, in this context, means the kind of ultrasound application, i.e., the specific harmonic imaging technique used (see above). This can be done either with a constant setting during the examination as well as with varying, e.g., the mechanical index (MI) in the course of the examination in order to achieve specific effects on the course of received (“reflected”) noise. Data processing, on the other hand, means the kind of analysis of the expected course of received noise alterations followed by specific US-CE application (either as a bolus application or as a constant infusion) according to the applied harmonic imaging regimen.

First reports were based on second harmonic imaging following a single application of US-CE (bolus kinetics) [1,2,4,5,6,7]. Depth of insonation was initially restricted to 10 cm due to technical constraints, i.e., only one hemisphere of the brain could be analyzed by the time (later called the “unilateral” approach). Received time intensity curves (TIC) were analyzed by dedicated algorithms, which derive specific parameters of wash-in and wash-out (such as time-to-peak intensity, TPI) by fitting the actual information (TIC) to the expected course defined by pre-described mathematical model functions [15] (compare Figure 5). Subsequent studies initially analyzed different harmonic imaging modes like phase inversion harmonic imaging (PIHI) [8] and also adapted “destructive” modes (applying higher MI) with the aim to increase signal-to-noise ratio (CBI and TVI) [10,17]. Due to the unilateral character of the examination, only qualitative information was extracted, i.e., perfusion could be classified as either normal or constricted. Another technical constraint of the unilateral approach is the fact that tissue close to the probe cannot be analyzed due to nearfield artifacts. Therefore, cortical areas of the brain cannot be evaluated.

Further technical approaches intended to extract qualitative information (i.e., the degree of perfusion restriction) by applying different acquisition and processing approaches. The refill kinetics approach applied a combination of low MI and high MI imaging during a constant infusion of contrast enhancer [18]. The hypothesis was to destroy the US-CE by an ultra-quick series of high MI pulses and then to display the “refilling” of tissue perfusion by low MI imaging, which should be dependent on the state of perfusion. A given algorithm extracts specific parameters, which have been proven to represent semi-quantitative parameters in myocardial perfusion imaging. A different approach was to apply a longer series of relatively slow frequent and high MI pulses during a constant infusion of US-CE and thereby to evaluate the “depletion” of tissue perfusion, which should also be dependent on the state of perfusion (CODIM) [9]. Figure 5 displays considerations on the mathematical function describing three theoretical courses of time intensity curves of different kinetic models.

Another attempt to extract (semi-) quantitative data was introduced as the so-called bilateral approach [8]. Here, imaging depth was set to 15 cm, visualizing not only one but both hemispheres in one examination (compare Figure 2 and Figure 3). This became possible due to improved ultrasound machines and the introduction of second generation US-CEs (Optison^®^, SonoVue^®^), improving signal-to-noise ratio. Two potential advantages were claimed. First, utilizing the so-called mirror approach, intra-individual comparison of perfusion parameters in both affected and unaffected hemispheres could facilitate semi-quantitative analyses. A prerequisite would be the depth-independence of at least one relevant parameter, which could especially be proven for the time dependent parameter, time-to-peak intensity (TPI) [11]. Second, once the affected hemisphere was on the far side of the probe, cortical areas of the affected hemisphere could also be evaluated for perfusion impairments. Since cortical areas are frequently involved in territorial infarction, this was seen as a relevant improvement.

Irrespective of data acquisition and processing modality, the evaluation of specific parameters can be performed two-fold, either by the analysis of pre-defined regions of interest (ROI) or by the presentation of parametric images, where data analysis is carried out by pixel-wise presentation according to one specific parameter (e.g., time-to-peak intensity) [8]. Both processing modalities are offered by industrial providers by now and have been tested against dedicated solutions recently [13]. Figure 6 displays both ROI-wise analysis and a parametric image in an acute stroke patient.

## 3. Validation to Standard Imaging

Validating different UPI approaches to standard imaging has been crucial from the beginning. In the early studies, patients presenting with ischemic strokes were evaluated in a sub-acute time window up to 24–48 h after symptom onset [1,2,4,5,6,7]. Therefore, the actual target was the identification of already infarcted tissue, which was tested mainly against follow-up, non-contrast CCT. Since the focus of interest shifted toward the differentiation between ischemic and penumbral tissue, validation tools needed to become more sophisticated. However, CT (or MRI) perfusion imaging has not always been as well accessible as it is today. Hence, one approach was to define parenchymal tissue as normal, delayed-, or not-perfused in the acute UPI examination and correlate this classification to infarcted and non-infarcted tissue in follow-up CCT according to early clinical course [12]. The hypothesis was that both delayed- and not-perfused tissue of initial UPI should be infarcted in follow-up CCT once there had not been clinical improvement in the meantime. Once there had been distinctive clinical improvement, only not-perfused tissue of the initial UPI exam should be infarcted in the follow-up CCT.

As a matter of fact, especially CT perfusion imaging gained a lot of interest at that time and started its impressive road of success not just in clinical stroke medicine. Nowadays, CT perfusion imaging is widely accessible, probably being the most important factor why the significance of UPI has not further evolved. However, later UPI studies employed timely, correlated CT or MRI perfusion imaging and recently proved that the bilateral approach of high MI bolus imaging, in particular, could distinguish between unimpaired, delayed, and nullified perfusion [19]. Pre specified ROIs in both hemispheres were determined; TPI values of the unaffected hemisphere served as an intra-individual normal value. Values of the affected hemisphere yielded the perfusion status, either for specified ROIs or displayed as parameter image for the whole imaging plane. Once TPI was within ±4 s as compared with the intra-individual normal value, perfusion was unimpaired; a delay of more than 4 s indicated critically hypo-perfused tissue, and nullified rise of TIC indicated infarction. Hereby, the ability of the method to detect penumbral tissue in acute stroke was claimed.

## 4. Clinical Applications up to Date and Future Indications

Most of the UPI studies have been so far performed in acute stroke patients as described above. Besides contrast-enhanced imaging of cerebral vessels, UPI has already been mentioned in the EFSUMB guidelines and recommendations on the clinical practice of contrast-enhanced ultrasound (CEUS) in 2012 [20]. Studies have mainly been performed in territorial infarction due to main vessel occlusion. One study proved that infarctions as small as 2 cm in diameter can be reliably detected [21]. Case series have demonstrated detectable perfusion impairments in non-occlusive diseases as well [22,23]. In these applications, the bilateral approach utilizing a high MI setting following a bolus application of contrast enhancer seems to deliver the most robust information on the clinical questioning, focusing on vessel occlusion and penumbral imaging. Future challenges of UPI in acute stroke should focus on multicenter validation of up-to-date study results as well as the potential of mobile application. First attempts of mobile cerebral ultrasound imaging in acute stroke have focused on vessel imaging [24], but also basic perfusion imaging is challenged in one industrial project [25]. In addition to being a bedside method, UPI may also be used for serial studies in order to follow-up on brain perfusion. One indication may be early detection of successful recanalization. Serial assessment of UPI may also be used for the guidance of hemodynamic therapy to optimize cerebral perfusion with UPI as a surrogate marker. In a small study in stroke patients, improvement of cerebral perfusion detected by UPI was achieved due to systemic hemodynamic optimization [26].

However, different indications may require different technical settings. Whilst ischemic stroke remains the domain of UPI, it also has been used for identifying different acute or subacute cerebral lesions other than ischemic. There are a few studies on patients with intracranial hemorrhage (ICH) where UPI was used either to improve sonographic detectability of ICH or to describe perihemorrhagic penumbral perfusion (compare Figure 7). ICH can be detected as a hyperechogenic mass lesion within the brain parenchyma with a high sensitivity and specificity [27]. Detection and especially clear distinction of ICH from the adjacent tissue may be difficult in severe cerebral microangiopathy, in lobar hemorrhage, or in only small lesions. Comparable to CT-perfusion studies with a recess or severe hypo-perfusion of contrast media within the hemorrhagic lesion [28], UPI shows a recess of ultrasound contrast media especially within the ICH core and massive reduction of contrast media within the hemorrhagic lesion. Consecutively, ICH appears hypo-echogenic compared to the adjacent tissue, which is perfused normally as shown by the contrast agent with a clear delineation of the border of ICH from the surrounding tissue. Thus, detection of ICH volume may be improved significantly, especially in serial measurements [29]. Despite perihemorrhagic edema, the area of hypo-perfusion or non-perfusion in ICH is fairly restricted to the hemorrhagic lesion itself with no or a very narrow area of hypo-perfused tissue, e.g., perifocal penumbral perfusion. Conversely, parenchymal hemorrhagic transformation of ischemic stroke due to early spontaneous recanalization is difficult to distinguish from primary ICH on native scan but is characterized by a significantly larger perifocal penumbral zone of hypo-perfused tissue exceeding the hemorrhagic lesion by far [30]. Thus, UPI not only helps delineating the border of ICH for more valid volume measurement but also allows distinction of primary ICH from PHI.

Even though bedside monitoring of cerebral perfusion in brain trauma and acute or chronic subdural hematoma is extremely interesting and theoretically may help in guiding therapy—for instance by defining a surgical need in chronic SDH by detection of cortical hypo-perfusion due to venous compromise—studies on UPI are lacking and data on brain perfusion in these patients generally are scarce. Another application of UPI currently under scientific evaluation is the setting of aneurysmal subarachnoid hemorrhage (NCT02907879). UPI is evaluated with respect to its potential to diagnose cerebral hypo-perfusion in the course of cerebral vasospasm.

Various authors have evaluated cerebral tumors and their ultrasound perfusion patterns [31,32,33,34,35]. UPI is not only able to increase the differentiation of normal brain tissue from brain tumors but it is also helpful to differentiate different tumor types according to their perfusion pattern [32]. Tumor tissue shows a dramatic rise of contrast enhancement and high peak intensities compared to normal brain parenchyma [32]. When comparing benign and malignant tumors, there were no significant differences in peak intensities of the time–intensity curves, yet malignant tumors showed shorter times-to-peak intensities [32]. In the eyes of the authors, UPI is a rapid, practical and cost-effective technique, especially in critically ill patients or if multiple consecutive examinations are necessary. During intraoperative application, ultrasound allows the surgeon to localize a lesion in real-time even before the opening of the dura. This facilitates the surgical access and is a useful add-on to neuronavigation [36,37]. In addition, UPI enables the surgeon to assess tumor enhancement, vascularity, and perfusion, and to control for completeness of resection [38,39,40]. UPI has been applied in a variety of different brain tumors, e.g., gliomas and metastases [40,41]. In a recent study, Prada et al. characterized intraoperative contrast-enhanced ultrasound images of various brain tumors [40]. They also found a high accuracy between US-based real-time neuronavigation and preoperative MRI findings. The authors concluded that contrast application is useful for the localization, definition of borders, and depiction of the vascularization and perfusion pattern of brain tumors [40]. In another study, UPI was specifically evaluated in brain space-occupying lesions and could identify specific patterns of brain perfusion [42]. It could be shown that meningeomas and glioblastomas, if no large areas of necrosis were present, showed an increased perfusion, while in tumors with necrosis the perfusion was reduced as compared to normal tissue, although in total only 15 brain tumors were evaluated. In another study, it was shown that the differentiation between tumor and normal brain tissue was superior after administration of US-CE [41]. US-CE also enabled the control of completeness of resection, yet this was dependent on technical aspects like the position of the resection cavity. UPI has the potential to become a helpful tool for the surgeon during intraoperative application, yet larger studies are needed.

## 5. Restrictions of the Method and Safety Considerations

Despite the proven evidence of reproducibility and robustness, especially of time-based parameters of the bolus kinetic, no widespread application of UPI modalities has yet been achieved. Partly, this may be due to some well-known limitations of the method. First, a sonolucent transtemporal bone window is needed. Up to 15%–20% of the elderly patients present with an insufficient bone window, so that UPI is not applicable. Second, patients need to be compliant, so that the transducer can be held in position for the 45–60 s of data acquisition. Especially, severely affected patients may be agitated and therefore unsuitable for the method, bearing in mind that the procedure is hand-held. Third, using the bolus kinetic approach only one two-dimensional imaging plane can be evaluated per bolus application. Therefore, quantification is restricted to an investigation plane that has to be chosen beforehand. However, future development of three-dimensional insonation systems may overcome these limitations. Fourth, quantification is yet only semi-quantitative, i.e., no absolute values can be determined. However, quantification (in acute stroke) as described above utilizes the mirror approach, which is also common in CT and MRI perfusion imaging. In addition, quantification has only been proven for one parameter (TPI). Other parameters have to be challenged in future studies. Regarding safety of UPI, there have been apprehensions of side effects of both US-CE and administration of ultrasound pulses on brain integrity. These have mainly been triggered by results of studies applying long-lasting, whole brain, low-frequency insonation in the setting of ultrasound-enhanced thrombolysis, resulting in massive hemorrhage and blood–brain barrier disruption [43,44]. However, applying standard settings of transcranial insonation, UPI is regarded safe with no evidence of blood–brain barrier affection [45,46].

## 6. Conclusions

Cerebral ultrasound perfusion imaging has the potential to serve as a supplementary tool to conventional diagnostics in various clinical questionings. As long as temporal bone window is present, a multi-modal approach of vascular imaging for the detection of vessel occlusion, microvascular perfusion impairment or intracerebral hemorrhage is covered by the method. In addition, conventional contrast-enhanced imaging omitting the quantification of perfusion may serve as an extension of diagnostic properties. The unique feature of mobility facilitates application at the bedside. This could enable pre-hospital diagnostics, but also easy-to-apply follow-up diagnostics in the intensive care unit or stroke unit as well as in the operating room. Future developments should focus on multi-center studies to validate the findings described in this manuscript and the development of automated algorithms for examiner independence.

## Figures and Tables

**Figure 1 jcm-09-00816-f001:**
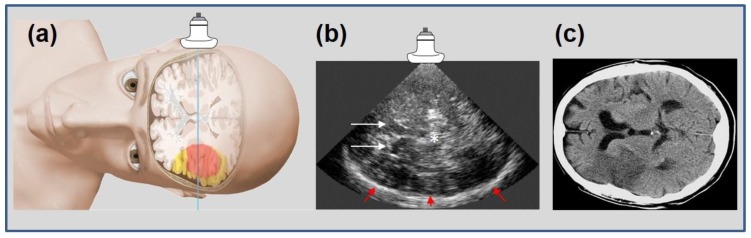
Schematic representation of insonation plane in transtemporal ultrasound imaging (**a**), adapted from [13] (with permission of copyright owner) with a corresponding “bilateral” B-mode image (**b**) with explanatory anatomical landmarks: white arrows = frontal horns of side ventricles; * = midline, third ventricle; red arrows = contralateral skull. Infarcted areas cannot be displayed in B-mode ultrasound. For orientation, comparison to conventional cerebral computed tomography scan, CCT, (**c**) with plane shifted by 90° accordingly, with an infarction in the hemisphere “contralateral” to the probe.

**Figure 2 jcm-09-00816-f002:**
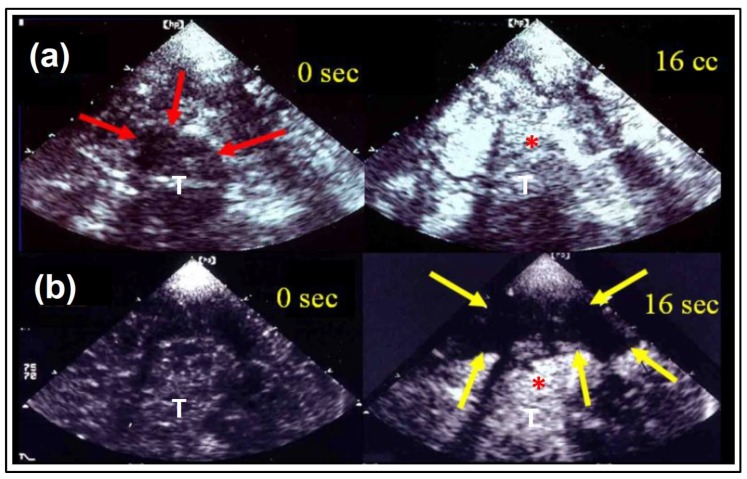
Ultrasound perfusion imaging in the course of time. Early “unilateral” gray-scale imaging in a healthy volunteer (**a**) and an acute stroke patient (**b**) corresponding to Figure 1: at baseline (0 s) and after contrast enhancer application at the time of maximal contrast enhancement (16 s). The thalamic region (red arrows) is marked with increase of brightness in both examples (*), whereas the regions of the lentiform nucleus and temporoparietal lobe are spared (yellow arrows), where later infarction was demonstrated in CCT follow-up. (T) indicates the third ventricle, adapted from [15] with permission from Elsevier.

**Figure 3 jcm-09-00816-f003:**
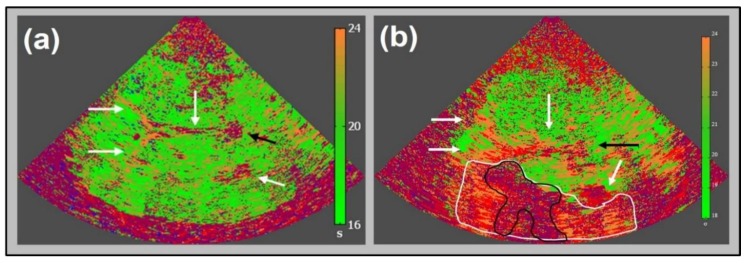
Up to date “bilateral” perfusion imaging corresponding to Figure 1: parametric image of time-to-peak intensity (TPI) in a healthy volunteer (**a**) with homogeneously distributed greenish parts of parenchymal structures (TPI 16 to 20 s) and the depiction of frontal and posterior horns of side ventricles as well as third ventricle (white arrows), and pineal gland (black arrow), adapted from [12]. Note the near field artifact. TPI parameter image of an acute stroke patient 2.5 h after symptom onset (**b**) of a severe stroke caused by occlusion of the M1 segment of the middle cerebral artery. Note the core of infarction (pink area, surrounded by black line) and hypo-perfused nature, and potentially salvageable area (orange area, surrounded by white line) due to collateral flow; adapted from [12] with the publisher’s permission.

**Figure 4 jcm-09-00816-f004:**
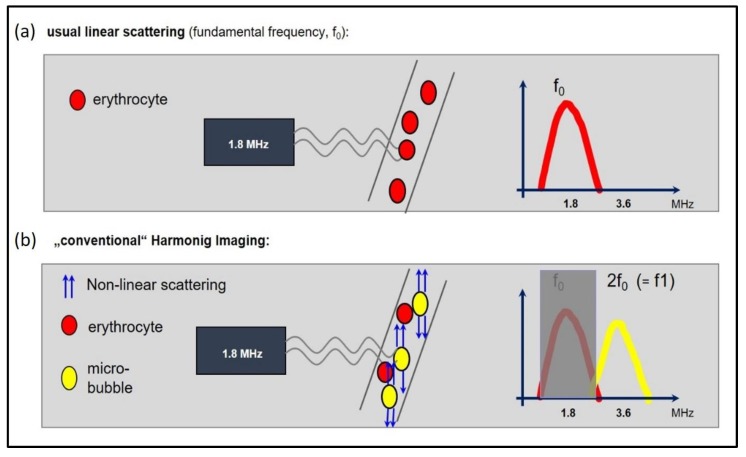
The basic principle of harmonic imaging. (**a**) When an ultrasound wave passes through tissue, the predominantly linear scattering of the erythrocytes results in a frequency, which is reflected back to the probe, which is equal to the transmitted frequency (here: fundamental frequency of 1.8 MHz). (**b**) “Harmonic imaging” due to non-linear scattering of the microbubbles: the resonance frequency of the microbubbles is typically a multiple of the transmitted (or fundamental) frequency. The harmonic frequencies are sent back to the probe, where they are used to create the image. Specifically, the second harmonic frequency (2f0) is used. The fundamental component is filtered out, so that that the received frequency of 3.6 MHz is two-fold higher than the transmitted frequency of 1.8 MHz.

**Figure 5 jcm-09-00816-f005:**
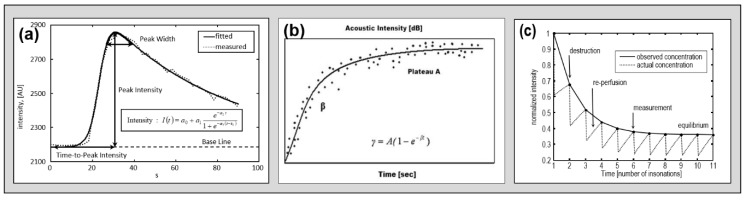
Theoretical course-of-time intensity curves in three different kinetic models as measured in models. Dotted lines represent measured concentration and straight lines represent course of fitted model function. (**a**) Bolus kinetic [8], (**b**) refill kinetic [18], and (**c**) depletion kinetic [15] with permission of the original publishers.

**Figure 6 jcm-09-00816-f006:**
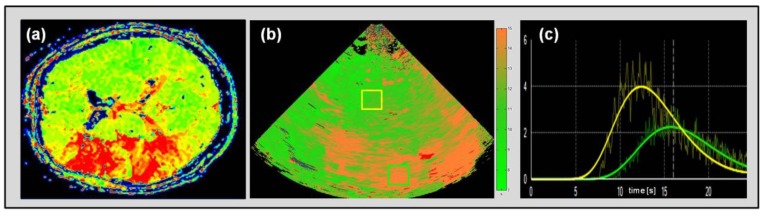
Perfusion MRI time-to-peak (TTP) map of an acute stroke patient with expanded penumbral perfusion delay in the territory of the middle cerebral artery (MCA) omitting basal ganglia (**a**). Ultrasonic cerebral perfusion imaging UPI parametric image with a corresponding depiction of time-to-peak intensity (TPI) delay in the MCA territory omitting basal ganglia (**b**). Exemplary depiction of ROI-wise course-of-time intensity curve in normal perfused brain tissue (yellow curve corresponding yellow box in (**c**) in basal ganglia of the unaffected hemisphere) and penumbral tissue (green curve and box) in an acute stroke patient with MCA occlusion and apparent collateral compensation.

**Figure 7 jcm-09-00816-f007:**
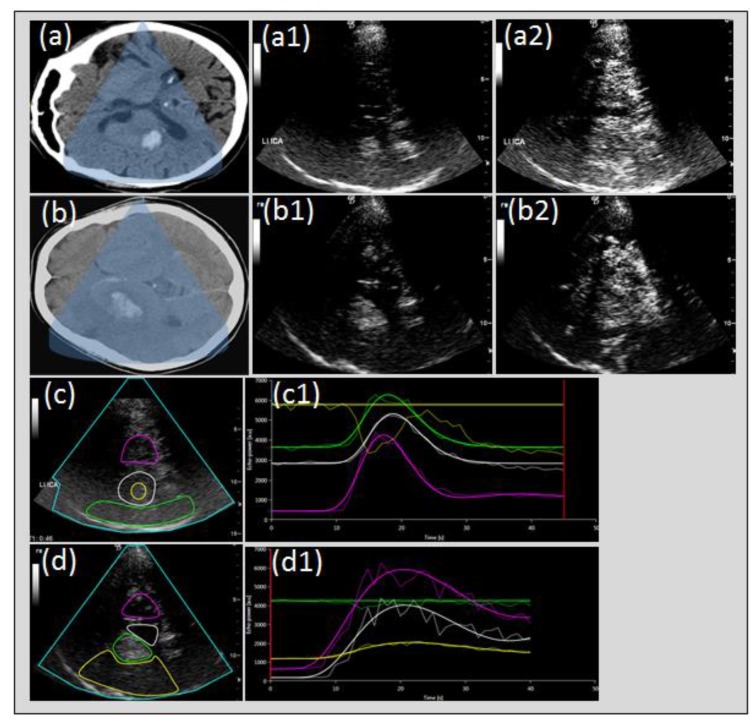
Perfusion imaging in intracerebral hemorrhage and hemorrhagic transformation of cerebral infarction. Cerebral CT of intracerebral hemorrhage (ICH) of the right basal ganglia, (**a**) native transcranial gray-scale sonography with hyperechogenic depiction of ICH (**a1**) and UPI with relative hypo-echogenicity of ICH compared to contrast perfusion of cerebral tissue (**a2**) due to non-perfusion constricted to the hemorrhagic lesion (**c**,**c1**). Cerebral CT of ICH due to hemorrhagic transformation (**b**), native transcranial gray-scale sonography with hyperechogenic depiction of hemorrhagic transformation (**b1**) and UPI with persistent hyperechogenicity of the hemorrhagic lesion due to omitted perfusion of the surrounding tissue due to acute stroke (**b2**) with slowed or missing tissue perfusion (**b2**,**d**,**d1**).

**Table 1 jcm-09-00816-t001:** Ultrasound contrast enhancers having been used in brain perfusion studies (adapted from [16]).

Name	First Approved	Gas	Shell Material	Producer/Distributor
Levovist^®^	1993, withdrawn	Air	Galactose microparticles	Schering AG, Berlin, DE
Optison^®^	1998	Octafluoropropane, C_3_F_8_	Cross-linked serum albumin	GE healthcare, Buckinghamshire, UK
SonoVue^®^	2001	Sulphurhexafluoride, SF_6_	Phospholipid	Bracco diagnostics, Milano, Italy

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
