# Peer review of "Twenty Years of Cerebral Ultrasound Perfusion Imaging—Is the Best yet to Come?"

_jcm, 2020, doi:10.3390/jcm9030816_

Round 1

Reviewer 1 Report

This review article entitled “20 years of cerebral ultrasound perfusion imaging –the best is yet to come?” is a narrative review describing the evolution of ultrasonic cerebral perfusion imaging (UPI) over the last 20 years and discussing its advantages and potential role in the future given its mobile applicability. The article is divided into: introduction, technical aspects, validation to standard imaging, clinical applications, and safety.

Overall, it is a well-organized summary of this literature. The topic is both relevant and interesting to readers given the current need for novel mobile imaging modalities in the pre-hospital setting for stroke care. However, it is difficult to read at times because of the need for English grammar/style editing. It would need thorough editing to be reconsidered.

To highlight just a few specific examples:

Abstract: “Validate results to golden standard” should read “gold” standard.

Abstract: “Since the identification of intracerebral hematoma and large vessel occlusion are also proven, UPI is a supplementary multi-modal imaging technique with the potential of preclinical application.” “Proven” is an awkward word to use in this sentence. Do the authors mean that ultrasound can reveal these findings? Furthermore, “preclinical” is a term usually used  to described laboratory research. The authors mean “pre-hospital” or “in field” application.

Introduction: “two of them being the mobile bedside character of the examination and the missing radiation exposure.” Perhaps “absence of radiation exposure” would be more appropriate.

Introduction: While the imaging modality is portable, expert image acquisition and interpretation may not be portable. One still needs a qualified person to obtain and interpret the images. This should be discussed as a potential limitation.

Introduction: “In neurosonology, acute ischemic stroke is the leading indication for ultrasound imaging.” I would recommend re-phrasing this because in most centers ultrasound is not used acutely for thrombolysis or thrombectomy decisions, but is used for working up and following carotid stenosis in the subacute and long term settings.

Figure 2: Probably better to describe “healthy volunteer” instead of “normal person”.

Page 4, Line 110: Delete comma between “defined” and “cerebral imaging”.

Page 6, Line 175: PIHI was already defined in previous section.

Page 7, Line 198: “so-called” is used here but also very frequently throughout the manuscript. Perhaps just a style preference, but it may best to avoid this.

Page 8, Line 241: “perfuzed” should be “perfused”

Page 9, Line 265: “One study could proof that infarctions as small as 2 cm…” This should read “one study proved”.

There are many other minor grammar/style edits throughout that are required in my opinion.

Reviewer 2 Report

The pertinence of the manuscript is justified by the fact that there is a lack of reviews about these issues for a long time. 

It is not clear the unilateral and bilateral transcranial approach. This could be important for understanding the following figures. A shorte sentence explaining theses approaches would be grateful 

Figure 1

In this figure, the head CT infarct zone does not seem to match that of the ultrasound B-mode. Please change this or explain better

CCT - please do not abbreviate

Figure 2 

NIHSS - 13 is irrelevant - omit this

Figure 3 -omit and apparent collateral compensation (not measured) 

Resolution of figures 5 is not enough and formulas are blurred 

Please explain - is this measured in the transcranial US? another organ? selected ROI? experimental? healthy?

Needs update

Figure 7 - each figure in this panel must have a corresponding letter to understand more clearly the legend of this figure. 

"In the early studies, patients presenting with ischemic strokes were evaluated in a sub-acute time 235 window up to 24-48 hours after symptom onset" There's no reference

The last three subtitles must be improved

  • there's an overuse of non-scientific language like "the idea is..." 
  • a more concise description of the data
  • the are no good validation studies and this must be more clearly addressed. 
  • Despite this, authors frequently use "UPI can be used..." which disagrees with the paucity of data validating this approach
  • a 2-column table with a list of pathologies and the basic information that UPI could get would be helpful to catch the readers attention
  • A figure/table with a proposed protocol  would be much useful for the reader

Round 2

Reviewer 1 Report

A native English speaker should give this a final round of copy editing. No further concerns in terms of content.